# Chromosome conformation capture resolved near complete genome assembly of broomcorn millet

Junpeng Shi [1], Xuxu Ma[1], Jihong Zhang[1], Yingsi Zhou[1], Minxuan Liu[2], Liangliang Huang[1], Silong Sun [1], Xiangbo Zhang[1], Xiang Gao[1], Wei Zhan[3], Pinghua Li[4], Lun Wang[5], Ping Lu[2], Haiming Zhao[1], Weibin Song[1] & Jinsheng Lai [1,6]

Broomcorn millet (*Panicum miliaceum* L.) has strong tolerance to abiotic stresses, and is probably one of the oldest crops, with its earliest cultivation that dated back to *ca.* ~10,000 years. We report here its genome assembly through a combination of PacBio sequencing, BioNano, and Hi-C (in vivo) mapping. The 18 super scaffolds cover ~95.6% of the estimated genome (~887.8 Mb). There are 63,671 protein-coding genes annotated in this tetraploid genome. About ~86.2% of the syntenic genes in foxtail millet have two homologous copies in broomcorn millet, indicating rare gene loss after tetraploidization in broomcorn millet. Phylogenetic analysis reveals that broomcorn millet and foxtail millet diverged around ~13.1 Million years ago (Mya), while the lineage specific tetraploidization of broomcorn millet may be happened within ~5.91 million years. The genome is not only beneficial for the genome assisted breeding of broomcorn millet, but also an important resource for other *Panicum* species.

---

[1] State Key Laboratory of Agrobiotechnology and National Maize Improvement Center, Department of Plant Genetics and Breeding, China Agricultural University, Beijing 100193, P. R. China. [2] Institute of Crop Science, Chinese Academy of Agricultural Sciences, Beijing 100081, P. R. China. [3] Annoroad Gene Technology (Beijing) Co., Ltd, Beijing 100176, P. R. China. [4] State Key Laboratory of Crop Biology, College of Agronomy, Shandong Agricultural University, Tai'an 271018, P. R. China. [5] Institute of Crop Germplasm Resources, Shanxi Academy of Agricultural Sciences, Taiyuan 030031, P. R. China. [6] Center for Crop Functional Genomics and Molecular Breeding, China Agricultural University, Beijing 100193, P. R. China. These authors contributed equally: Junpeng Shi, Xuxu Ma, Jihong Zhang. Correspondence and requests for materials should be addressed to W.S. (email: songweibin@cau.edu.cn) or to J.L. (email: jlai@cau.edu.cn)

Millets are a variety of small-seeded grass that are widely grown, particularly in resource-poor areas around Asia and Africa[1]. Broomcorn millet (Panicum miliaceum L.), also known as common millet or proso millet, is probably one of the oldest crops around the world, with its origin from northern China that could be dated back to ca. ~10,000 years before present (cal yr BP)[2]. Archeological evidence also suggested another possible origin center of broomcorn millet in eastern Europe, with its earliest cultivation that occurred ~7000 cal yr BP[3,4]. Broomcorn millet was a recent allotetraploid (2n = 4 × = 36) with its two subgenomes originated from two species closely related to P. capillare and P. repens[3,5]. It could be used as a pioneer crop at marginal regions due to its short growing season (reaching maturity after ~60–90 days), extremely low water requirements, high salt tolerance, and nutrient resource usage efficiency[6,7]. However, the grain yield of broomcorn millet is relatively low when compared with its close relative foxtail millet, which may be partially attributed to its little genetic gains up to now. As a demonstration of the feasibility of using 10X linked reads to assemble highly complex crop genomes, a draft genome sequence of broomcorn millet has been recently released[5]. However, no high-quality reference genome has been reported in the entire Panicum genus, which includes other important species, such as switchgrass (Panicum virgatum L.)[8].

The genomes of crop plants often have undergone polyploidization and have relatively high proportion of repeat elements (especially LTR retrotransposons)[9,10]. Despite the prosperity of assembling crop genomes due to the application of Illumina sequencing technologies, the majority of assemblies were remained to be in draft status due to the difficulty of assembling short Illumina reads[11]. Owing to the advent of SMRT (Single Molecule Real-Time) sequencing technologies, especially the popularization of PacBio sequencing which could generate reads up to ~30–40 kb, the continuity of genome assemblies (such as rice[12,13], maize[14,15], Aegilops tauschii[16,17], durian[18], and quinoa[19,20]) reached several orders of magnitude higher (N50 > 1 Mb) when compared with Illumina assemblies (usually tens of kilobases). In addition, the combination of BioNano optical mapping and chromosome conformation information, generated by either in vivo (Hi-C)[18,21] or in vitro (Chicago)[19] technologies, was shown to be able to anchor the scaffolds into chromosomal or subchromosomal levels.

Here, we report a high-quality de novo genome assembly of broomcorn millet. Through a combination of PacBio sequencing, BioNano optical mapping and Hi-C (in vivo) mapping, we generate a chromosome scale assembly with a total scaffold length of ~848.4 Mb (N50 of ~8.24 Mb) that accounted for ~95.6% of the estimated genome size (~887.8 Mb). We annotate 63,671 genes in broomcorn millet, which is nearly two times the number of genes (34,584) in foxtail millet. There are 19,609 genes in foxtail millet that are syntenic with broomcorn millet, among which 16,884 (~86.2%) genes have two homologous copies retained in broomcorn millet, indicating rare gene loss after whole-genome duplication (WGD). Phylogenetic analysis reveals a common ancestor before ~13.1 million years ago (Mya) between broomcorn millet and foxtail millet, and the tetraploidization is estimated to be happened within ~5.91 million years. The genome sequence we report here is not only important to understand the dynamic evolution following genome tetraploidization in Paniceae, but also benefit the molecular breeding of broomcorn millet in the future.

## Results

**Genome sequencing and assembly of broomcorn millet**. To sequence the genome of broomcorn millet, we selected an elite cultivar named Longmi4 (Supplementary Fig. 1), which was widely cultivated among the northern region of China. We first generated ~103.4 Gb Illumina paired-end reads (150 bp) to analyze the genome of Longmi4, which was highly homozygous (the heterozygosity ratio was ~0.04%)[22]. The genome size was estimated to be ~887.8 Mb from k-mer analysis (Supplementary Fig. 2 and Supplementary Note 1). It was about two times the size of foxtail millet (~485 Mb)[23,24], which was consistent with previous assumption that a tetraploidization event happened in the lineage of broomcorn millet after its divergence with Setaria[23]. While, there was a large difference of genome size between broomcorn millet and switchgrass (~1220.0 Mb, Panicum virgatum v1.0, DOE-JGI, http://phytozome.jgi.doe.gov/), since two independent polyploidization events were inferred to be happened in these two species[23].

We totally generated ~150.7 Gb subreads (N50 = 12.6 kb) from PacBio Sequel platform, which covered ~170 x of Longmi4 genome (Supplementary Table 1). Falcon[25] was firstly used to self-correct and assemble the PacBio reads, then polished with both PacBio and Illumina reads (~116 ×) to generate 1262 consensus contigs (~839.0 Mb) with contig N50 of ~2.55 Mb (Supplementary Note 2). We further generated BioNano optical maps (~235 ×, N50 = 255.2 kb, Supplementary Table 2), resolved the conflicts in original contigs into 1308 contigs and anchored these contigs into 905 scaffolds (~848.4 Mb) with scaffold N50 of ~8.24 Mb (Table 1). About ~95.6% of the estimated genome was covered by the scaffolds, with 127 longest scaffolds that accounted for more than 90% of the genome (N90 = 1.47 Mb). To evaluate the assembly quality, we mapped both the Illumina and RNA-seq reads back to the scaffolds, with mapping efficiencies of ~99.6% and ~91.5%, respectively (Supplementary Table 3). We also evaluated the assembly with 1440 Benchmarking Universal Single Copy Orthologs (BUSCO) genes from embryophyta[26], of which 1417 genes (~98.4%) were annotated and 1411 genes (~98.0%) were intact.

To further anchor and orient the scaffolds into super scaffolds, we constructed Hi-C (in vivo cross-linking of chromatins) libraries from the seedlings and generated ~622.2 million paired-end reads covering ~140.2 x of Longmi4 genome (Supplementary Table 4). As expected, the spatial proximity, as reflected by the Hi-C interaction intensity, decreased along with

**Table 1 The statistics of genome assembly and protein-coding genes in broomcorn millet**

| | Contigs | Scaffolds | Super scaffolds |
|---|---|---|---|
| *Assembly features* | | | |
| Numbers | 1308 | 905 | 475 |
| Total length | ~838.9 Mb | ~848.4 Mb | ~848.4 Mb |
| N50 | ~2.55 Mb | ~8.24 Mb | ~48.26 Mb |
| Longest | 19,200,716 bp | 22,633,379 bp | 69,183,459 bp |
| Coverage | ~94.5% | ~95.6% | ~95.6% |
| GC content | ~46.80% | | |
| *Protein-coding genes* | | | |
| Numbers of genes | 63,671 | | |
| Gene space (coverage) | 214,072,228 bp (~24.1%) | | |
| Number of transcripts | 86,387 | | |
| Mean transcript length | ~2883 bp | | |
| Mean CDS length | ~1023 bp | | |
| Mean intron length | ~1270 bp | | |

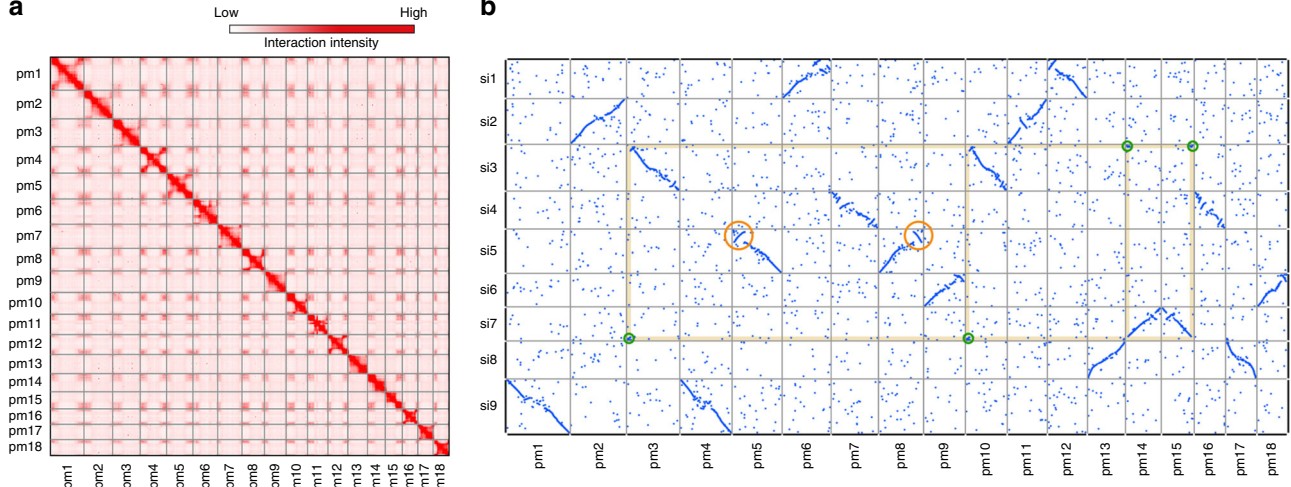

**Fig. 1** The Hi-C assisted assembly of Longmi4 pseudomolecules. **a** Heatmap showing Hi-C interactions under the resolution of 200 kb, and the anti-diagonal pattern for the intrachromosomal interactions may reflect the Rabl configuration of chromatins. **b** Genome comparison between broomcorn millet (pm1–pm18) and foxtail millet (si1–si9). Each dot represented a homologous sequence reported by Mummer, and the two orange circles referred to two intrachromosomal inversions. The green circles represented the quartet of interchromosomal exchanges that may be happened before tetraploidization. Source Data of Fig. 1b are provided as a Source Data file

the increasing of physical distance between two loci (Supplementary Fig. 3). Therefore, we were able to cluster and orient 444 scaffolds (838.9 Mb, ~98.9%) into 18 long super scaffolds (hereafter denoted as pseudomolecules, N50 = 48.3 Mb) through a hierarchical clustering strategy[27]. Upon the pseudomolecules we have constructed (Supplementary Table 5), the Hi-C interaction matrices displayed a distinct anti-diagonal pattern for the intrachromosomal interactions (Fig. 1a), which may reflect the so-called Rabl configuration that the long and short arms of chromatins folded parallelly in the interphase nuclei[21,28].

Comparative genomic analysis revealed a 2-to-1 syntenic relationship between broomcorn millet and foxtail millet (Fig. 1b). Unlike maize which experienced strong chromosome fusions after tetraploidization[29], the two sets of chromosomes were nearly intact in broomcorn millet. However, intrachromosomal rearrangements, especially inversions, were pervasive between these two genomes. For example, two large inversions (~11.8 Mb and ~8.9 Mb) were identified on the two homologous chromosomes of Longmi4 (Pm5 and Pm8) that supported by intact scaffolds (Fig. 1b and Supplementary Fig. 4). Besides, we also identified an interchromosomal exchange which was estimated to be occurred before the tetraploidization event, as reflected by a quartet among the chromosome ends of Pm3, Pm10, Pm14, and Pm15.

**Gene annotation and gene family analysis**. To annotate the protein-coding genes in broomcorn millet, we generated RNA-seq data from the aerial part of Longmi4 seedlings and collected public available RNA-seq data of broomcorn millet, which covered the major developmental stages and tissues with a total data volume of ~68.6 Gb (Supplementary Table 3). We also used the protein sequences from other related plant species, including *S. italica*[23,24], *P. glaucum*[30], *S. bicolor*[31], *Z. mays*[15], *O. sativa*[32], and *A. thaliana*[33] to perform homology-based prediction (see Methods). In total, we predicted 63,671 genes in broomcorn millet, which was nearly two times the number of genes in *S. italica* reference genome Yugu1 (34,584 genes, v2.2)[23]. There were 62,934 genes (~98.8%) that could be assigned to 18 pseudomolecules, with the gene density highly skewed toward the distal ends of chromosome arms (Supplementary Fig. 5). The average length of transcripts (~2883 bp), coding regions (~1023 bp), and introns (~1270 bp) in broomcorn millet were highly similar with

other important cereal crops (Supplementary Table 6). As a consequence of WGD, the percentage of WGD or segmental duplicated genes in broomcorn millet (39,769 genes, ~63.2%) was substantially higher than foxtail millet (5805 genes, ~16.9%). On the contrary, only 5248 genes (~8.3%) in broomcorn millet were identified as singletons, which were much lower than that in foxtail millet (7367 genes, ~21.5%, Supplementary Table 7).

Using foxtail millet as a representative of the genome organization of two ancestral diploid genomes in broomcorn millet, we are able to study the gene loss and retentions following tetraploidization in broomcorn millet[29]. We identified 19,609 genes in foxtail millet (~56.7% of total genes in Yugu1) that were syntenic with at least one subgenome in broomcorn millet (Supplementary Table 8). In consistent with a nearly double number of genes in broomcorn millet (63,671) compared with foxtail millet (34,584), we found the majority (16,884, ~86.2%) of syntenic genes in foxtail millet have two homologous copies retained in broomcorn millet (Fig. 2 and Supplementary Table 8). The remaining 2725 (~13.8%) genes in foxtail millet have only one syntenic homolog in broomcorn millet. It was contrasting with the drastic gene loss after WGD reported in maize[29] and soybean[34], which may be due to the more recent WGD in broomcorn millet as compared with soybean and maize (discussed later). Also, the two subgenomes of broomcorn millet shown approximately the same level of gene retentions (Fig. 2), as opposed to the bias of gene fractionation in maize[29] and *Brassica rapa*[35].

We further annotated the genes in Longmi4 via the homology of functional domains from InterPro database[36]. In total, there were 62,270 genes (~97.8%) that could be annotated by InterProScan, among which 46,299 (~72.7%), 53,289 (~83.7%), 19,959 (~31.3%), and 41,091 (~64.5%) genes could be annotated by Pfam[37], PANTHER[38], ProSite[39], and Gene3D[40], respectively (Supplementary Fig. 6). We assigned Gene Ontology (GO) to 35,560 genes (~55.8%) which included biological process (N = 21,517), cellular component (N = 8216), and molecular functions (N = 31,667). We also identified 3273 transcription factors from 55 gene families in Longmi4, including the major families of MYB, bHLH, ERF, NAC, and bZIP which contained 366, 295, 293, 243, and 171 genes, respectively (Supplementary Table 9).

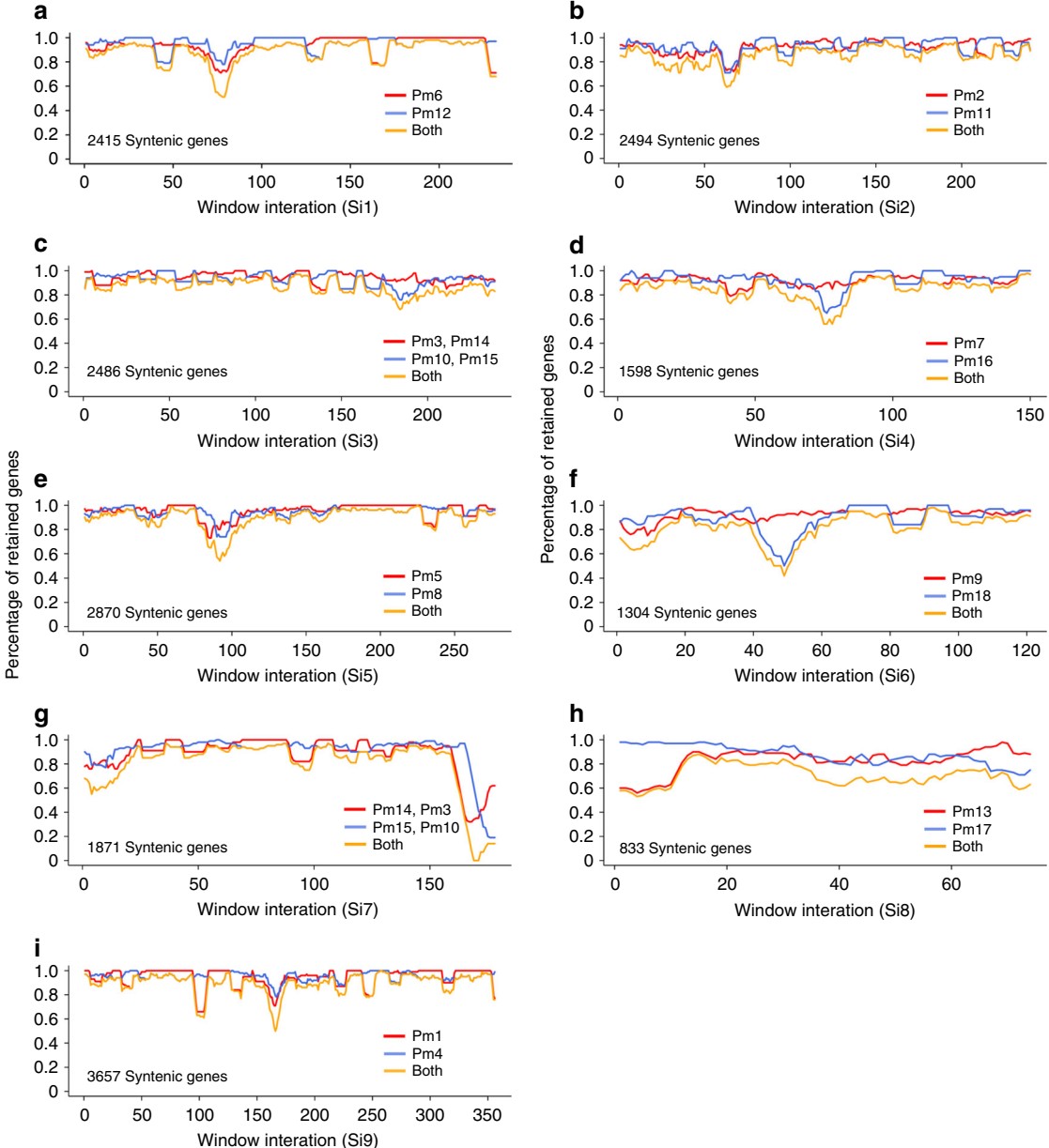

**Fig. 2** Gene loss and retentions in broomcorn millet. **a** sliding window approach with window size of 100 syntenic genes and step size of 10 syntenic genes was used to show the percentage of retained genes in subgenome1 (red), subgenome2 (blue), and both (yellow) in broomcorn millet using foxtail millet (a~i) as a reference. In total, there were 19,609 genes in foxtail millet that were syntenic with at least one subgenome in broomcorn millet, among which 16,884 (~86.2%) syntenic genes have two homologous copies retained in broomcorn millet. Source Data are provided as a Source Data file

To further reveal the lineage-specific expansion of genes families, we clustered the genes of broomcorn millet with foxtail millet, pearl millet, sorghum, and maize by OrthoMCL[41], and identified 12,022, 11,686, 10,776, 11,399, and 11,104 families for each species, respectively (Supplementary Fig. 7). As expected, within the majority (7124, ~83.6%) of shared gene families (8517), the numbers of genes in broomcorn millet were higher than that in both foxtail millet and pearl millet. Furthermore, the numbers of lineage-specific gene families ($N = 801$, Supplementary Fig. 7) and genes (6592) therein were both highest in broomcorn millet, some of which may be due to the neofunctionalization after WGD[42]. We annotated the lineage-specific genes in broomcorn millet and found that it contained a variety of functional domains, including Cytochrome P450, NB-ARC domains (also called NBS domains), sugar transporters, and

wall-associated receptor kinase, etc. GO enrichment analysis of lineage-specific genes revealed 13 significant terms in molecular functions, mainly involved in hydrolase activities (Supplementary Fig. 8).

Mining the genome of broomcorn millet uncovered the genes potentially involved in both biotic and abiotic stress resistance in broomcorn millet. In total, we identified 493 genes containing NB-ARC domain (Pfam: PF00931) that may be involved in disease resistance[43], of which 20 genes (seven gene families) were specific in broomcorn millet. In consistent with that in pearl millet[30], the distribution of NB-ARC genes was highly biased in broomcorn millet, with gene clusters observed at chromosome ends of pm13 and pm17 (Supplementary Fig. 9). As a crop with extremely strong drought tolerance, we also identified 15 ABA or WDS (water-deficiency stress) responsive genes (Pfam: PF02496)[44] in

broomcorn millet. Interestingly, four of these ABA genes were constitutively expressed with relatively high expressional level across all the samples we examined, even for the tissues or stages without salt or drought treatment (Supplementary Fig. 10). Further experimental validations are needed to elucidate the functional role of these genes in the drought tolerance of broomcorn millet.

**Recent bursts of *Gypsy* elements in broomcorn millet**. By applying a de novo repeat family identification approach[45], we totally identified ~458.9 Mb sequence as repeat elements in broomcorn millet, which constituted ~54.1% of the genome. The percentage of repeat elements in broomcorn millet was between that of foxtail millet (~46.8%) and pearl millet (~68.0%, Supplementary Table 10). For the DNA transposons, the proportion of broomcorn millet (~4.8%) was comparable with pearl millet (~4.7%) while much lower compared with foxtail millet (~10.2%), including the superfamilies of *CMC-EnSpm*, *MULE-MuDR*, and *PIF-Harbinger*. Considering the close relationship among these three species, we inferred that lineage-specific bursts of DNA transposons may have been happened in foxtail millet. *Helitrons* were relatively low in broomcorn millet (~0.44%), foxtail millet (~0.63%), and pearl millet (~0.11%), as compared with that in maize (~2.2%)[46–48]. We also found a relatively even distribution for both DNA transposons and *Helitrons* along the chromosomes, except several relative depletions around centromeric regions (as defined by gene density valleys in the middle of chromosomes, Supplementary Fig. 5).

Similar to other cereal crops[23,30], LTR retrotransposon (~37.1%), especially the *Gypsy* superfamily (~31.4%), constituted the majority of repeat elements in broomcorn millet. *Gypsy* elements were highly enriched around centromeric regions in both broomcorn millet and foxtail millet. While, the distribution of *Copia* elements was contrasting between these two species, especially around the centromeric regions (Supplementary Figure 5). There were also large differences of *Gypsy*-to-*Copia* ratio among the major crops in Paniceae, with the highest of ~7.16 in broomcorn millet, followed by foxtail millet (~3.9), sorghum (~3.7), pearl millet (~2.24), and lowest in maize (~2.0). To explain these differences, we dated the activity of both *Gypsy* and *Copia* elements in these crops. We found very recent bursts of *Gypsy* elements in both maize and sorghum, followed by the bursts of other three species that were all within ~1 Mya (Fig. 3a). The amplifications of *Copia* elements were also very recent in foxtail millet, sorghum, and maize (<1 Mya), which was consistent with a previous estimation of LTR amplifications in these species[23,49]. While, the activity of *Copia* elements was relatively old in broomcorn millet (~2 Mya, Fig. 3b), which may explain the extremely high *Gypsy*-to-*Copia* ratio in broomcorn millet, since no recent bursts of *Copia* elements were detected.

**The phylogeny of Paniceae**. Previous studies have established the phylogeny in the grass subfamily Panicoideae, although the genetic information may be limited to have an exact estimation of the evolutionary timeline[23,50]. Furthermore, as a model species in Paniceae, the timing of the tetraploidization in the lineage of broomcorn millet remained unresolved. By taking advantage of the high-quality assembly of broomcorn millet in this study, in combination with the newly published genomes of pearl millet[30] and *Dichanthelium oligosanthes*[51], we were able to reconstruct the phylogeny in Paniceae (Fig. 4). We firstly estimated the *Ks* (synonymous substitution rate) of orthologous gene pairs between broomcorn millet and foxtail millet, and the peak of *Ks* (~0.162) nearly coincided with the peak between sorghum and maize (~0.152), which was consistent with the inference that the

lineages between broomcorn millet and foxtail millet diverged ~13.1 Mya[23], slightly earlier than that between sorghum and maize (~11.9 Mya)[52]. Phylogenetic data supported *Dichanthelium* as a distinct genus in Paniceae[51], while the peak of *Ks* between *Dichanthelium oligosanthes* and foxtail millet (~0.177, ~14.3 Mya) nearly colocalized with that between broomcorn millet and foxtail millet, indicating a close split of the progenitors among these three species (Fig. 4a). The divergence between foxtail millet and pearl millet was more recent, with the peak of *Ks* (~0.121) corresponded to ~9.81 Mya, slightly older than a previous estimation of ~8.3 Mya[23]. Finally, the Andropogoneae and Paniceae shared a common ancestor before ~23.5 Mya, as revealed by the peak of *Ks* between foxtail millet and sorghum (~0.286) and that between broomcorn millet and sorghum (~0.295).

We next estimated the time of tetraploidization in the lineage of broomcorn millet by calculating the *Ks* between paralogous genes. We found two peaks for the *Ks* of broomcorn millet, with the secondary one (~1.08, ~85.9 Mya) that far preceded the common ancestor between Andropogoneae and Paniceae, which may be originated from the WGD shared by all the grass as reported to be happened ~70 Mya[24]. We further confirmed this WGD in foxtail millet, since only a single peak of *Ks* was detected (~1.00, Fig. 4a) and no recent WGD happened in its lineage (Fig. 4b). We further estimated that the two ancestor genomes in broomcorn millet diverged ~5.91 Mya (*Ks* ~0.073), suggesting that the tetraploidization of broomcorn millet was more recent than that in soybean (*G. soja*, ~13 Mya)[34], probably also recent than maize (~5–12 Mya)[52]. In allotetraploid cottons (*G. hirsutum*), the divergence time between A- and D-progenitor genomes was estimated to be ~6.0–6.3 Mya, and the allotetraploid was formed around 1–1.5 Mya[53]. In addition, gene loss in both A and D genomes of tetraploid cottons were rare[53]. We thus hypothesized that the time of allotetraploid was more recent than ~5.91 Mya in broomcorn millet.

**Discussion**
We report here a high-quality reference genome of broomcorn millet. The quality of our assembly as reflected by the contig N50 (~2.55 Mb), scaffold N50 (~8.24 Mb), or super scaffolds were considerably better than several other recently accomplished crop genome assemblies (such as rice[12,13], maize[14,15], quinoa[19,20], barley[21], and durian[18]), which might be attributed to the unique combination of methods we used (deep PacBio sequencing, BioNano optical mapping, and in vivo Hi-C scaffolding), in addition to the fact that the genome of broomcorn millet is highly homozygous. Recently, both the in vivo Hi-C[21] and in vitro Chicago technology[18,19] have been shown to be able to order and orient the scaffolds into chromosomal level in several plant and animal species. We demonstrated here that a high-quality chromosome scale assembly could be generated through in vivo Hi-C mapping in a rather complex plant genome. Since in vivo Hi-C could also provide extra information of high-order chromatin architecture, it may potentially become more attractive as compared with the in vitro Chicago technology.

Unlike maize which experienced strong genome rearrangements and expansions after tetraploidization[52], broomcorn millet retained the majority of two copies of ancestral genes, kept the basal chromosome numbers ($2n = 4 \times = 36$) and experienced relatively weak genome expansion (as reflected by a nearly doubled genome size compared with foxtail millet), likely resulted from its more recent tetraploidization (within ~5.91 million years). No biased fractionation of duplicated genes was detected between two subgenomes of broomcorn millet, as opposed to the strong fractionation in maize that subgenome1 consistently retained more genes than subgenome2[29]. It was also interesting

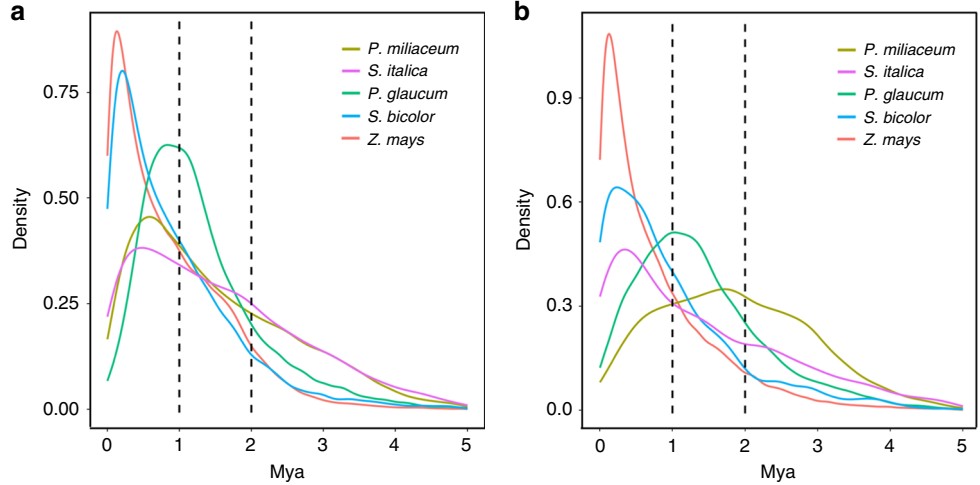

**Fig. 3** The activity of LTR retrotransposons in Paniceae. The insertion time of *Gypsy* (**a**) and *Copia* (**b**) superfamilies were calculated in broomcorn millet (*P. miliaceum*), foxtail millet (*S. italica*), pearl millet (*P. glaucum*), sorghum (*S. bicolor*), and maize (*Z. mays*), respectively

| | Peak 1 | Peak 2 |
|---|---|---|
| *S. italica - P. miliaceum* | 0.162 | – |
| *Z. mays - S. bicolor* | 0.152 | – |
| *S. italica - P. glaucum* | 0.121 | – |
| *D. oligosanthes - S. italica* | 0.177 | – |
| *S. italica - S. bicolor* | 0.286 | – |
| *P. miliaceum - S. bicolor* | 0.296 | – |
| *P. miliaceum - P. miliaceum* | 0.073 | 1.080 |
| *S. italica - S. italica* | – | 1.003 |

**Fig. 4** The phylogeny of Paniceae. **a** *Ks* distribution of orthologous and paralogous genes in Paniceae. We further confirmed two WGD events in foxtail millet (**b**) and broomcorn millet (**c**), with the paralogous genes in green circles referred to the WGD shared by all grass, and the paralogous genes in orange circles referred to lineage-specific tetraploidization in broomcorn millet. **d** The phylogeny of Paniceae inferred from *Ks* distributions. The timeline was calculated based on the divergence between sorghum and maize (*Ks* ~ 0.152, 11.9 Mya). The WGD events were marked by stars. Source Data of Fig. 4a are provided as a Source Data file

that the time between ~12 and ~15 Mya was an important period during the evolution of Paniceae, since a close split of the progenitors of broomcorn millet, foxtail millet, and *Dichanthelium oligosanthes* happened at that time interval (as well as that between sorghum and maize).

The availability of this genome will no doubt facilitate the comparative genomic researches between *Panicum* and other crops. Interestingly, as a tetraploid crop, the biomass of broomcorn millet is unexpectedly low, especially when compared with its close relative energy crop switchgrass. It was inferred that two independent tetraploidization events happened in these two close relatives[23]. Further comparative genomic analysis will eventually uncover the genetic basis of phenotype difference between these two closely related tetraploid species in the future.

## Methods

**Plant materials**. The seeds of Longmi4 were grown at dark conditions under 25 °C after sowing for 14 days, then the aerial parts of seedlings were harvested and mixed, frozen immediately in liquid nitrogen for the extraction of genomic DNA. High molecular genomic DNA was extracted from isolated nuclei for sequencing library construction.

**PacBio and Illumina sequencing**. Libraries for single molecule real-time (SMRT) PacBio genome sequencing were constructed following the standard protocols of Pacific Biosciences company. Briefly, high molecule genomic DNA was sheared to ~20 kb targeted size, followed by damage repair and end repair, blunt-end adaptor ligation, and size selection. Finally, the libraries were sequenced on the PacBio Sequel platforms.

The Illumina libraries were constructed according to the standard manufacturer's protocol (Illumina). Briefly, around 5 μg DNA was fragmented and followed by size selection (~450 bp) by agarose gel electrophoresis. The ends of selected DNA fragments were blunted with an A base overhang and ligated to sequencing adapters. All the libraries were sequenced on Illumina X-ten platform with pair-end sequencing strategy.

**Genome survey**. K-mer distribution was estimated by using jellyfish (http://www.genome.umd.edu/jellyfish.html, v2.2.6) with parameters -m 17 -s 200 M -C. The heterozygosity ratio was estimated by the online tool of GenomeScope (http://qb.cshl.edu/genomescope/). Finally, the genome size was calculated according to the formula that Genome_Size = K-mer coverage/Mean k-mer depth.

**De novo assembly and polish of the genome**. The raw contigs were assembled by Falcon (https://github.com/PacificBiosciences/falcon, v1.8.7)[25] with the following steps: (a) raw reads overlapping for error correction; (b) pre-assembly and error correction; (c) overlap detection and filtering; (d) graph construction and contigs generation. The parameters during Falcon assembly were listed as follows: length_cutoff = 11 Kb, length_cutoff_pr = 15 Kb, pa_HPCdaligner_option = -v -B128 -M24 -t12 -e 0.75 -k18 -w8 -h180 -T32 -l2800 -s1000, ovlp_HPCdaligner_option = -v -B128 -t12 -h280 -e 0.96 -k22 -T32 -l3200 -s1000. Then, the PacBio reads were mapped back to the raw contigs by Blasr (https://github.com/PacificBiosciences/blasr, v5.1)[54] with parameters (--bam --bestn 5 --minMatch 18 --nproc 4 --minSubreadLength 1000 --minAlnLength 500 --minPctSimilarity 70 --minPctAccuracy 70 --hitPolicy randombest --randomSeed 1) and contigs were further corrected by Arrow (https://github.com/PacificBiosciences/GenomicConsensus, v2.1.0) with the parameter -j 30. Finally, Illumina reads were mapped back to the improved contigs and corrected by Pilon (https://github.com/broadinstitute/pilon, v1.20)[55] to generate the final consensus contigs with parameters --genome reference.fasta --changes --vcf --diploid --fix bases --threads 40 --mindepth 20.

**BioNano optical maps and scaffold construction**. High molecular genomic DNA were digested with single-stranded nicking endonuclease Nt. BspQI, then labeled using the IrysPrep Labelling mix and Taq polymerase according to standard BioNano protocols. Labeled DNA was imaged with BioNano Irys system, and raw BioNano data were assembled into optical map by IrysSolve (https://bionanogenomics.com/support/software-downloads/, BioNano Genomics) with default parameters. Next, the optical map was aligned to PacBio contigs by IrysSolve, resolved the conflicts, and built the scaffolds according to the overlapping information between contigs and optimal maps.

**Hi-C library constructions**. The aerial parts of Longmi4 seedlings (14 days) were harvested and crosslinked by 40 ml 2% formaldehyde solution at room temperature for 15 min. Then, a total of 4.324 ml of 2.5 M glycine was added to quench the cross-linking reaction. The supernatant was removed and tissues were ground with liquid nitrogen and resuspended with 25 ml of extraction

buffer I (0.4 M sucrose, 10 mM Tris-HCl, pH 8.0, 10 mM MgCl₂, 5 mM β-mercaptoethanol, 0.1 mM phenylmethylsulfonyl fluoride [PMSF], and 1x protease inhibitor, Roche), then filtered through miracloth (Calbiochem). The filtrate was centrifuged at 3500*g* at 4°C for 20 min. The pellet was resuspended in 1 ml of extraction II (0.25 M sucrose, 10 mM Tris-HCl, pH 8, 10 mM MgCl₂, 1% Triton X-100, 5 mM β-mercaptoethanol, 0.1 mM PMSF, and 1x protease inhibitor) and centrifuged at 18,400*g* and 4 °C for 10 min. The pellet was resuspended in 300 μl of extraction buffer III (1.7 M sucrose, 10 mM Tris-HCl, pH 8, 0.15% Triton X-100, 2 mM MgCl₂, 5 mM β-mercaptoethanol, 0.1 mM PMSF, and 1 x protease inhibitor) and loaded on top of an equal amount of clean extraction buffer III, then centrifuged at 18,400*g* for 10 min. Discard the supernatant and then wash the pellet twice by resuspending it in 500 μl ice cold 1x CutSmart buffer and then centrifuging the sample for 5 min at 2500*g*. The nuclei were washed by 0.5 ml of 1 x restriction enzyme buffer and transferred to a safe-lock tube. Next, the chromatin is solubilized with dilute SDS and incubation at 65 °C for 10 min. After quenching the SDS by Triton X-100, overnight digestion was applied with 4 bp cutter restriction enzyme (400 units MboI) at 37 °C on rocking platform. The next steps are Hi-C-specific, including marking the DNA ends with biotin-14-dCTP and performing blunt-end ligation of cross-linked fragments. The proximal chromatin DNA was re-ligated by ligation enzyme. The nuclear complexes were reversed crosslinked by incubating with proteinase K at 65 °C. DNA was purified by phenol–chloroform extraction. Biotin-C was removed from non-ligated fragment ends using T4 DNA polymerase. Fragments were sheared to a size of 100–500 base pairs by sonication. The fragment ends were repaired by the mixture of T4 DNA polymerase, T4 polynucleotide kinase, and Klenow DNA polymerase. Biotin-labeled Hi-C sample were specifically enriched using streptavidin magnetic beads. The fragment ends were adding A-tailing by Klenow (exo-) and then adding Illumina paired-end sequencing adapter by ligation mix. Finally, the Hi-C libraries were amplified by 10–12 cycles PCR, and sequenced in Illumina HiSeq instrument with 2 × 150 bp reads.

**Pseudomolecule construction by Hi-C**. Clean Hi-C reads were aligned to scaffolds using bowtie2 (v2.0.5)[56] with end-to-end model. Totally, we generated ~622.2 million pair-end reads and ~115.6 million were uniquely aligned to the scaffolds (Supplementary Table 3). Reads with low mapping quality (maq < 20), multiple hits, duplications, and singletons were discarded. Then, HiC-Pro (https://github.com/nservant/HiC-Pro, v2.7.8)[57] was used to detect the ligation site using an exact matching procedure and to align the 5′ end of reads back to the scaffolds. There were ~64.9 million valid interaction pairs that were used to build the interaction matrices and draw the heatmap with Juicebox (https://github.com/aidenlab/Juicebox, v1.8.8) software[58]. We then used Lachesis (https://github.com/shendurelab/LACHESIS)[27] to cluster, order, and orient the scaffolds. First, we clustered 495 scaffolds (839,447,339 bp, ~98.95%) into 18 chromosome groups according to the agglomerative hierarchical clustering algorithm. Within each cluster, a minimum spanning tree is found and the longest path in the tree is extracted as the trunk, an incomplete but high-confidence ordering of scaffolds within each chromosome group[27]. There were 170 scaffolds that were ordered as trunks, which constituted 808,559,470 bp (~95.3%) of the total scaffolds. Scaffolds were excluded from the trunks were reinserted between the trunks that maximized the amount of linkage between adjacent scaffolds, which resulted in 444 scaffolds that can be ordered (838,831,330 bp, ~98.9%). Finally, for each chromosome cluster, we ordered and traversed all the direction of the scaffolds through a weighted directed acyclic graph (WDGA) to predict orientation of each scaffold.

**Gene annotation and gene family analysis-**. To annotate the protein-coding genes in broomcorn millet, we combined three approaches: ab initio prediction, RNA-seq, and protein homology-based predictions. For the protein homology-based prediction, we downloaded the protein sequences of *S. italica*, *P. glaucum*, *S. bicolor*, *Z. mays*, *O. sativa*, and *A. thaliana* from Phytozome (http://www.phytozome.net), then aligned to the assembled scaffolds with TBLASTN (*e*-value < 1e-5). Alignments within 20 kb were merged, and the alignments with coverage >85% and identity >75% were remained. Next, Genewise was used to annotate the gene models according to the alignments. For the RNA-seq based prediction, we generated RNA-seq data for the aerial parts of seedlings of Longmi4 grown at 25 °C after sowing for 14 days, combined with the publicly available RNA-seq data (NCBI SRA accessions: ERR2040773, SRR1697309, SRR1697310, SRR2179899~SRR2179908, SRR2179952, SRR2179961, SRR4069168~SRR4069173, Supplementary Table 2). All the RNA-seq data were aligned to scaffolds with Tophat2[59], and transcripts were further assembled by Cufflinks[60] and StringTie[61]. Transdecoder was then used to predict the ORFs, and ORFs with length shorter than 600 bp were filtered if no protein homology information (from TBLASTN alignments) or Pfam[37] domains from HMMer[62] (*e*-value < 1e-5) could support this ORF. For the ab initio prediction, the repeat sequence in scaffolds were firstly masked by Repeatmasker (http://repeatmasker.org/, open-4.0.7), then the repeat masked scaffolds were annotated by Fgenesh[63]. The genes models from ab initio prediction will be retained if: (a) the predicted protein sequence could be aligned to the protein database above by BLASTP; (b) the predicted protein sequence could be supported by Pfam[37] database (*e*-value < 1e-5); (c) more than 100 RNA-seq reads

covered the coding region of the ab initio gene models. Finally, considering a better prediction of ORF structure for RNA-seq and homology-based predictions, we merged the gene models with the priority of RNA-seq-based > homology-based > ab initio with in-house Perl scripts. Potential transposons were further removed according to the protein annotation from InterProScan, which finally resulted in 63,671 protein-coding genes in Longmi4.

We used InterProScan (http://www.ebi.ac.uk/interpro/download.html, v5.15-54.0) to annotate the functional domains of genes with parameters -f tsv -goterms -iprlookup -dp -T. The transcription factors genes were annotated by the PlantTFDB (v4.0, http://planttfdb.cbi.pku.edu.cn/)[64]. GO enrichment analysis was performed by AgriGO (version 2, http://systemsbiology.cau.edu.cn/agriGOv2/). To cluster the genes into gene families, we aligned the representative protein sequence of each gene from broomcorn millet, foxtail millet, pearl millet, sorghum, and maize together by BLASTP (e-value < 1e-5), then clustered the genes by OrthoMCL (http://orthomcl.org/orthomcl/)[41] with e-value cutoff of −20 and percent match cutoff of 50.

**Analysis of gene loss and retentions**. To identify the duplicated genes in broomcorn millet, we aligned the representative protein of each gene from Longmi4 by blastp (-e 1e-10 -b 5 -v 5 -m 8 -o -a 8), then classified the type of duplicated genes by MCScanX (http://chibba.pgml.uga.edu/mcscan2/) with default parameters. To analyze the extent of gene loss and retentions in broomcorn millet, we used Coge pipeline (https://genomevolution.org/CoGe/) and blast the CDS of broomcorn millet against foxtail millet by Last, then used DAGchainer and QuotaAlign to find and merge syntenic blocks. Fractionation analysis was further applied by setting the syntenic depth to 2-to-1 between broomcorn millet and foxtail millet. A sliding window approach with window size of 100 syntenic genes and step size of 10 genes were used to show the proportion of retained genes in broomcorn millet.

**Identification of repeat elements**. We used a de novo repeat identification approach to annotate the repeat elements in broomcorn millet, foxtail millet[23], and pearl millet[30]. First, we used RepeatModeler (open-1.0.11) to train a repeat database by the NCBI blast approach (-engine ncbi), then annotated the repeat elements according to the build database above by RepeatMasker (http://repeatmasker.org/, open-4.0.7). To more accurately identity the LTR retrotransposons, we used LTRharvest[65] (http://genometools.org/, v1.5.9) to identify the candidate LTRs with the parameters: -v -mintsd 5 -maxtsd 20, then annotated the inner proteins of LTRs by LTRdigest with the parameters: -pptlen 10 30 -pbsoffset 0 3 -trnas -hmms. Candidate LTRs that were classified into Gypsy and Copia superfamilies were processed into activity analysis. We extracted sequences of the long terminal repeats for each LTR, aligned them with MUSCLE (v3.8.31), then calculate the distance $K$ with Kimura Two-Parameter approach between LTRs by distmat in EMBOSS (v6.6.0). The activity of each LTR was calculated by the formular: $T = K/(2 \times r)$, where $r$ refers to a general substitution rate of $1.3 \times 10^{-8}$ per site per year in grass family[66].

**Phylogeny of broomcorn millet**. We used the Coge pipeline (https://genomevolution.org/coge/) to perform the comparative genomic analysis between species in Paniceae. Briefly, we used Last to blast the CDS against others, then used DAGchainer and QuotaAlign to find and merge syntenic blocks. We set the expected syntenic depth (1-to-1 or 2-to-1) to filter redundancy, then calculated the $Ks$ rate between orthologous genes by CodeML. The $Ks$ of paralogous genes were calculated by a similar approach. The time of the common ancestor between species was inferred according to the $Ks$ of ~0.152 between maize and sorghum, since maize and sorghum were inferred to have a common ancestor before ~11.9 Mya[52].

**Code availability**. All the custom codes in this study were available upon request or can be downloaded from GitHub (https://github.com/caulai/broomcorn-millet-genome-assembly).

## Data availability

Genome assembly was deposited into NCBI Genbank with accession ID PPDP00000000. The genome assembly and annotations have also been deposited in the Genome Warehouse in BIG Data Center, Beijing Institute of Genomics (BIG), Chinese Academy of Sciences, under accession number GWHAAEZ00000000. The genome sequence and gene annotations could also be found at Coge with genome ID 50980. The transcriptome data, Illumina reads (resequencing and Hi-C) and PacBio bam files generated in this study were deposited into NCBI SRA with accession number SRP128667. The source data underlying Figs. 1b, 2, 4a, and Supplementary Figs 2, 6 are provided as a Source Data file. A reporting summary for this Article is available as a Supplementary Information file.

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

## Acknowledgements

We thank the NextOmics (Wuhan, China) for their assistance to generate the PacBio data and genome assembly. We also thank BerryGenomics (Beijing, China) for their assistance in generation of the BioNano data. We appreciated Dr. Shi, Yiting from China Agricultural University for her helpful discussion with this manuscript. This work was supported by the National Key Research & Development Program (2016YFD0101803, 2017YFD0101104, 2016YFD0100404), National Natural Science Foundation of China (31421005, 91635303) and 948 project (2016-X33).

## Author contributions

J.S., W.S., and J.L. designed this research project. J.S., J.Z., M.L., X.G., P.H.L., L.W., P.L., H.Z., W.S., and J.L. collected the plant materials, performed the experiments, and generated the sequencing data. J.S. and X.M. assembled the contigs and scaffolds. J.S., W.Z., and L.H. analyzed Hi-C data. J.S. and Y.Z. annotated the genome. J.S. and X.M. performed comparative and evolutionary analysis. X.Z. and S.S. participated in the data analysis. J.S. and J.L. wrote the paper.
