## [Peer Review File · Nature Communications]

Reviewers' comments:

Reviewer #1 (Remarks to the Author):

In the revision of the manuscript, Shi et al., make several changes that significantly improve the manuscript. However, some concerns still remain in the current version. Specifically, there still is no relevant biology associated with this high-quality genome assembly, i.e. C4 or drought, genes lost or retained or neofunctionalized during brief fractionation.

The HiC data significantly improves the assembly and illuminated previously concerns regarding using the setaria genome to resolve into chromosomes. Although it was mentioned in response to reviews, it would be good to validate the HiC orientation using the BioNano maps for the whole assembly not just the rearrangements.

Thank you for adding your genome to CoGe, which will be a great resource for the community. It also was important to asses some of the new concerns with the HiC based chromosome scale assembly.

Changing the name to Broomcorn Millet in the manuscript made it more consistent with literature.

The authors did not address the concerns of the reviewers regarding the ABA section. Almost all of the reviewers mentioned this section in one way or another as lacking evidence. This reviewer thinks that this section is completely misleading and is grounds for rejection of the manuscript if not removed or significantly changed.

Without any work on the C4 genes this manuscript is still just a very high-quality genome announcement. The authors miss a very unique opportunity to mine the high-quality assembly gene loss and retention after the recent WGD.

The authors need to remove all statements about being first. Some listed below

Line 42: "Being the first reference genome in the genus Panicum..."

line 78: "first reference genome in the genus of Panicum..."

Grammar is still an issue and several examples are highlighted below. This is not an exhaustive list.

Line 36-37: "accounted by 18 super scaffolds that each corresponding to one chromosome"

Line 38: "indicating rare gene loss compared with its closely related diploid crop, foxtail millet..."

Line 53: "could be served as a pioneer crop at marginal regions due to its..."

Line 64: "the majority of assemblies were remained to be in "draft""

Line 102: "since they were proposed to..."

Line 141: "exchange which was estimated to be happened before tetraploidization..."

Line 179: "In consistent with pearl..."

Line 218: "that lineage specific bursts of DNA transposons may have been happened in foxtail..."

Reviewer #2 (Remarks to the Author):

The authors have generated a genome assembly of Broomcorn Millet, the first reference in the genus *Panicum*. This resource will be useful for the genetics and breeding community.

Comments:

1. The authors need to provide an independent quality control of the genome sequence. Using a high-density genetic map would be a good.
2. In the section on drought tolerance the authors need to preface the section with something like, "To demonstrate the utility of the genome sequence ..." Although, I am not sure how this section adds to the paper.
3. In the abstract and results sections the authors state that there has been rare gene loss in broomcorn millet compared to the closely related foxtail millet. What is the evidence for this statement other than the total number of genes? In addition, on line 155 what is the % of homologous copies retained?
4. On line 96 it says that the Longmi genome was highly homozygous. What is the % homozygosity? Was Longmi inbred for multiple generations before sequencing?
5. On line 106-107, how do you polish a PacBio assembly with PacBio reads?
6. Line 274-275 I am not sure what is meant by "probably also recent than maize.."
7. There are numerous typos and awkwardly worded sentence that will need to be cleaned up.
8. Code availability is upon request, why not place it in Github?

Reviewer #3 (Remarks to the Author):

This manuscript describes the whole genome sequence and assembly and genome analysis of broomcorn millet. This species has undergone a recent WGD, is considered to be one of the earliest cultivated crops and is known to be resistant to abiotic stress. In terms of a genomics problem, assembly of this crop is notable because it has undergone a relatively recent whole genome duplication (WGD), which has the potential to confound assembly. Despite the WGD, the assembly seems to have proceeded smoothly. The authors contend that this whole genome sequence and assembly provides the first reference genome in the genus *Panicum*. Thus, the genome sequences generated in this study are beneficial for future genome assisted breeding of broomcorn millet (a relatively unimproved crop species), but also an important resource for other *Panicum* species. Additionally, the analysis of the proso millet genome could better establish the phylogeny of the Panicoidea and better resolve the timing of the WGD event.

In general, the authors have made strong attempts to address previous comments and concerns brought up by reviewers. In particular I note a significant improvement in describing the methods, as well as rectifying some confusion around numbers associated with assembly metrics.

There are two main concerns remaining with the manuscript:

Echoing previous reviewers, I note that the bulk of the manuscript presents the results of a routine sequence, assembly and annotation process. While the outcome is a resource that will have use within the plant science community, the methods to obtain it are well documented and not novel. Additionally, while there is some discussion involving drought responsive genes and phylogenetic placement, these do not substantially advance our understanding of broomcorn millet, especially with respect to its drought tolerance. Thus, the manuscript does lack novelty and immediate application to agricultural problems. While I think this is a nice resource for the plant community (as I mentioned previously), I don't see that it is of high enough impact for a NATURE publishers journal.

The written English is poor, and the manuscript suffers from a number of grammar issues. All previous reviewers noted this,

and the author's claim to have thoroughly revised the manuscript, but there remains much work to be done.

Reviewers' comments:

Reviewer #1 (Remarks to the Author):

In the revision of the manuscript, Shi et al., make several changes that significantly improve the manuscript. However, some concerns still remain in the current version. Specifically, there still is no relevant biology associated with this high-quality genome assembly, i.e. C4 or drought, genes lost or retained or neofunctionalized during brief fractionation.

Thank you for your time and comments. In our revised manuscript, we analyzed the gene losses and retentions by using foxtail millet as a representative of the genome organization of two ancestral diploid genomes in broomcorn millet, and found that ~86.2% of the syntenic genes in foxtail millet having two homologous copies retained in broomcorn millet, indicating rare gene loss after WGD in broomcorn millet (Line 163-177). In addition, the section related to ABA/drought tolerance was significantly toned down (Line 208-214).

The HiC data significantly improves the assembly and illuminated previously concerns regarding using the setaria genome to resolve into chromosomes. Although it was mentioned in response to reviews, it would be good to validate the HiC orientation using the BioNano maps for the whole assembly not just the rearrangements.

The BioNano maps could be used to anchor and orient contigs into scaffolds through the Hybrid Scaffolding process of BioNano Solve pipeline. During this process, there were 46 contigs with potential mis-joins that were resolved. The pseudomolecules were further constructed upon the scaffolds by using Hi-C, so it may be powerless to further validate the pseudomolecules by BioNano maps. We aligned the BioNano genome maps (N = 831, total length of 864.3 Mb) to the pseudomolecules constructed by Hi-C, and found that 811 (~98%) maps with a total length of 802

Mb (~95%) could be aligned, indicating a high consistency between BioNano maps and pseudomolecules.

Thank you for adding your genome to CoGe, which will be a great resource for the community. It also was important to assess some of the new concerns with the HiC based chromosome scale assembly.

Changing the name to Broomcorn Millet in the manuscript made it more consistent with literature.

The authors did not address the concerns of the reviewers regarding the ABA section. Almost all of the reviewers mentioned this section in one way or another as lacking evidence. This reviewer thinks that this section is completely misleading and is grounds for rejection of the manuscript if not removed or significantly changed.

The section related to ABA/drought tolerance was significantly toned down (Line 208-214).

Without any work on the C4 genes this manuscript is still just a very high-quality genome announcement. The authors miss a very unique opportunity to mine the high-quality assembly gene loss and retention after the recent WGD.

In our revised manuscript, we analyzed the gene loss and retentions after WGD in broomcorn millet using foxtail millet (Yugu1) as a diploid reference. We identified 19,609 genes in foxtail millet that were syntenic with at least one homologous in broomcorn millet, among which 16,884 (~86.2%) genes were syntenic with two homologous copies in broomcorn millet, indicating rare gene loss after WGD in broomcorn millet. Please see Line 163-177 for details.

The authors need to remove all statements about being first. Some listed below

Line 42: "Being the first reference genome in the genus Panicum..."

line 78: "first reference genome in the genus of Panicum..."

All the statements about being 'first' was removed in our revised manuscript.

Grammar is still an issue and several examples are highlighted below. This is not an exhaustive list.

Line 36-37: "accounted by 18 super scaffolds that each corresponding to one chromosome"

Line 38: "indicating rare gene loss compared with its closely related diploid crop, foxtail millet..."

Line 53: "could be served as a pioneer crop at marginal regions due to its..."

Line 64: "the majority of assemblies were remained to be in "draft""

Line 102: "since they were proposed to..."

Line 141: "exchange which was estimated to be happened before tetraploidization..."

Line 179: "In consistent with pearl..."

Line 218: "that lineage specific bursts of DNA transposons may have been happened in foxtail..."

Thank you for pointing out the grammar issues in our manuscript, we have checked and revised them thoroughly.

Reviewer #2 (Remarks to the Author):

The authors have generated a genome assembly of Broomcorn Millet, the first reference in the genus *Panicum*. This resource will be useful for the genetics and breeding community.

Thank you for your time and comments to our manuscript.

Comments:

1. The authors need to provide an independent quality control of the genome sequence. Using a high-density genetic map would be a good.

The combination of PacBio sequencing, BioNano optical mapping and Hi-C mapping was proved to be powerful in assembling large complex genomes such as Barley (Mascher, et al., 2017.), Quinoa (Jarvis, et al., 2017.) and Durian (The, et al., 2017.). In this study, we assembled the genome into 1,308 contigs and 905 scaffolds with fairly high quality (contig N50 ~2.55 Mb & scaffold N50 ~8.24 Mb & ~95% of the genome coverage). The high quality of scaffolds could also be proved by the high mapping rate (~98%) of BioNano genome maps and very few gaps (~9.5 Mb). The pseudomolecules were constructed upon the scaffolds by using Hi-C, and the high quality of scaffolds will no doubt enhance the accuracy of pseudomolecules, as evidenced by the Rabl configuration of chromatins, the synteny with foxtail millet and the chromosome distribution of protein coding genes.

The construction of genetic maps was laborious and time-consuming, and we are still working hard to build a F2 population in broomcorn millet. We used a genetic map (67,580 markers, unpublished) from Dr. Zhang Heng's lab to check the accuracy of our pseudomolecules. Over all, there were very high rate of consistency between our pseudomolecules and the genetic maps, with over 98% of the scaffolds' position and orientation perfectly aligned with genetic map, while only 7 of 444 scaffolds that anchored into pseudomolecules (~1.6%) showing some conflicts with genetic map.

2. In the section on drought tolerance the authors need to preface the section with something like, "To demonstrate the utility of the genome sequence ...". Although, I am not sure how this section adds to the paper.

The section related to ABA/drought tolerance was significantly toned down in our revised manuscript (Line 208-214).

3. In the abstract and results sections the authors state that there has been rare gene loss in broomcorn millet compared to the closely related foxtail millet. What is the evidence for this statement other than the total number of genes? In addition, on line 155 what is the % of homologous copies retained?

In our revised manuscript, we analyzed the gene loss and retentions after WGD in broomcorn millet using foxtail millet (Yugu1) as a diploid reference. We identified 19,609 genes in foxtail millet that were syntenic with at least one homologous in broomcorn millet, among which 16,884 (~86.2%) genes were syntenic with two homologous copies in broomcorn millet, indicating rare gene loss after WGD in broomcorn millet. Please see Line 163-177 for details.

4. On line 96 it says that the Longmi genome was highly homozygous. What is the % homozygosity? Was Longmi inbred for multiple generations before sequencing?

The heterozygosity ratio of Longmi4 genome was ~0.04% as estimated by GenomeScope (Line 96-97). The broomcorn millet was self-pollinated without the need for artificial selfing.

5. On line 106-107, how do you polish a PacBio assembly with PacBio reads?

The raw contigs was polished with PacBio reads by Arrow (<https://github.com/PacificBiosciences/GenomicConsensus>) which applied a more straightforward hidden Markov approach. Please see *De novo assembly and polish of the genome* in **Online Method** section.

6. Line 274-275 I am not sure what is meant by “probably also recent than maize..”

Maize is an allotetraploid with its two progenitors separated ~11.9 Mya, while the tetraploidization was inferred to be occurred before ~5 Mya (Swigonova, et al., 2004. & Schnable, et al., 2011.). We estimated the two ancestors of broomcorn millet diverged ~5.91 Mya, so the tetraploidization occurred after ~5.91 Mya. So, it has the possibility that the tetraploidization of broomcorn millet happened slightly earlier or at the same time with maize, although the analysis of gene loss and retentions suggested that it was very likely to be a recent allotetraploid.

7. There are numerous typos and awkwardly worded sentence that will need to be cleaned up.

We have revised our sentence thoroughly in revised manuscript.

8. Code availability is upon request, why not place it in Github?

Thank you for your suggestion. We have put our custom scripts into GitHub with the following URL (<https://github.com/caulai/broomcorn-millet-genome-assembly>).

Reviewer #3 (Remarks to the Author):

This manuscript describes the whole genome sequence and assembly and genome analysis of broomcorn millet. This species has undergone a recent WGD, is considered to be one of the earliest cultivated crops and is known to be resistant to abiotic stress. In terms of a genomics problem, assembly of this crop is notable because it has undergone a relatively recent whole genome duplication (WGD), which has the potential to confound assembly. Despite the WGD, the assembly seems to have proceeded smoothly. The authors contend that this whole genome sequence and assembly provides the first reference genome in the genus *Panicum*. Thus, the genome sequences generated in this study are beneficial for future genome assisted breeding of broomcorn millet (a relatively unimproved crop species), but also an important resource for other *Panicum* species. Additionally, the analysis of the proso millet genome could better establish the phylogeny of the Panicoidea and better resolve the timing of the WGD event.

Thank you for your time and comments to our work.

In general, the authors have made strong attempts to address previous comments and concerns brought up by reviewers. In particular I note a significant improvement in describing the methods, as well as rectifying some confusion around numbers associated with assembly metrics.

There are two main concerns remaining with the manuscript:

Echoing previous reviewers, I note that the bulk of the manuscript presents the results of a routine sequence, assembly and annotation process. While the outcome is a resource that will have use within the plant science community, the methods to obtain it are well documented and not novel. Additionally, while there is some discussion involving drought responsive genes and phylogenetic placement, these do not substantially advance our understanding of broomcorn millet, especially with respect to its drought tolerance. Thus, the manuscript does lack novelty and immediate application to agricultural problems. While I think this is a nice resource for the plant community (as I mentioned previously), I don't see that it is of high enough impact for a NATURE publishers journal.

Taken reviewers' suggestions, in our revised manuscript, we found strong gene retentions of WGD genes in broomcorn millet (~86.2% of syntenic genes in foxtail millet have two homolog copies in broomcorn millet), which added additional biological insights related to this high-quality crop genome which experienced relatively recent tetraploidization. The availability of this high-quality genome sequence will also facilitate the genome assisted breeding and comparative genomic analysis in other *Panicum* species.

The written English is poor, and the manuscript suffers from a number of grammar issues. All previous reviewers noted this, and the author's claim to have thoroughly revised the manuscript, but there remains much work to be done.

We have further checked the grammar thoroughly in revised manuscript.

REVIEWERS' COMMENTS:

Reviewer #1 (Remarks to the Author):

The authors have addressed my concerns.

Reviewer #1 (Remarks to the Author):

The authors have addressed my concerns.

Thank you very much for your time and critical comments that substantially improved our manuscript.